# Altered Serum Alpha1-Antitrypsin Protease Inhibition before and after Clinical Hematopoietic Stem Cell Transplantation: Association with Risk for Non-Relapse Mortality

**DOI:** 10.3390/ijms25010422

**Published:** 2023-12-28

**Authors:** Ido Brami, Tsila Zuckerman, Ron Ram, Batia Avni, Galit Peretz, Daniel Ostrovsky, Yotam Lior, Caroline Faour, Oisin McElvaney, Noel G. McElvaney, Eli C. Lewis

**Affiliations:** 1Department of Clinical Biochemistry and Pharmacology, Ben-Gurion University of the Negev, Be’er-Sheva 8410501, Israel; lewis@bgu.ac.il; 2Hematology Department and Bone Marrow Transplantation Unit, Rambam Health Care Campus, Haifa 3109601, Israel; t_zuckerman@rambam.health.gov.il; 3Bone Marrow Transplantation Unit, The Division of Hematology, Tel-Aviv Sourasky Medical Center, Tel-Aviv 6423906, Israel; ronr@tlvmc.gov.il; 4Department of Bone Marrow Transplantation and Cancer Immunotherapy, Hadassah-Hebrew University Medical Center, Ein Kerem, Jerusalem 9112001, Israel; batiaa@hadassah.org.il; 5Department of Hematology, Soroka University Medical Center, Be’er-Sheva 8410101, Israel; galitipa@gmail.com; 6Clinical Research Center, Soroka University Medical Center and Faculty of Health Sciences, Ben-Gurion University of the Negev, Be’er-Sheva 8410101, Israel; ostrdani@post.bgu.ac.il; 7Division of Anesthesiology, Pain and Intensive Care, Tel-Aviv Sourasky Medical Center, Tel-Aviv 6423906, Israel; matoyster@gmail.com; 8Ruth and Bruce Rappaport Faculty of Medicine, Technion, Israeli Institute of Technology, Haifa 3109601, Israel; carolinefaour1@gmail.com; 9The Irish Centre for Genetic Lung Disease, Royal College of Surgeons in Ireland, Beaumont Hospital, D02 YN77 Dublin, Ireland; oisinmcelvaney@rcsi.ie (O.M.); gmcelvaney@rcsi.ie (N.G.M.)

**Keywords:** acute-phase response, autoinflammation, checkpoint inhibitors, immunomodulation, wound healing

## Abstract

α1-Antitrypsin (AAT), an acute-phase reactant not unsimilar to C-reactive protein (CRP), is a serine protease inhibitor that harbors tissue-protective and immunomodulatory attributes. Its concentrations appropriately increase during conditions of extensive tissue injury, and it induces immune tolerance, in part, by inhibiting the enzymatic activity of the inflammatory serine protease, proteinase 3 (PR3). Typically administered to patients with genetic AAT deficiency, AAT treatment was recently shown to improve outcomes in patients with steroid-refractory graft-versus-host disease (GVHD). GVHD represents a grave outcome of allogeneic hematopoietic stem cell transplantation (HSCT), a potentially curative intervention for hematological diseases. The procedure requires radio/chemotherapy conditioning of the prospective marrow recipient, a cytotoxic process that causes vast tissue injury and, in some formats, interferes with liver production of AAT. To date, changes in the functional profile of AAT during allogeneic HSCT, and during the cytotoxic intervention that precedes HSCT, are unknown. The present study followed 53 patients scheduled for allogeneic HSCT (trial registration NCT03188601). Serum samples were tested before and after HSCT for AAT and CRP levels and for intrinsic anti-proteolytic activity. The ex vivo response to clinical-grade AAT was tested on circulating patient leukocytes and on a human epithelial cell line treated with patient sera in a gap closure assay. According to the ex vivo experiments, circulating leukocytes responded to AAT with a favorable immune-regulated profile, and epithelial gap closure was enhanced by AAT in sera from GVHD-free patients but not in sera from patients who developed GVHD. According to serum collected prior to HSCT, non-relapse mortality was reliably predicted by combining three components: AAT and CRP levels and serum anti-proteolytic activity. Taken together, HSCT outcomes are significantly affected by the anti-proteolytic function of circulating AAT, supporting early AAT augmentation therapy for allogeneic HSCT patients.

## 1. Introduction

Cytotoxic ablation protocols for the elimination of hematologic malignancies are performed prior to allogeneic hematopoietic stem cell transplantation (HSCT); however, graft-versus-host disease (GVHD) often ensues and is associated with substantial morbidity and mortality [1]. While eradicating leukemic cells, cytotoxic ablation protocols also cause widespread tissue damage, compromise the critical gastrointestinal barrier, and lead to excitatory interactions between antigen-presenting cells (APCs), such as dendritic cells (DCs), and T cells, setting the grounds for an unwanted, yet potent, alloimmune response [2,3,4].

Upon activation, DCs increase membrane CD40 levels and promote CD4^+^ T-cell and memory CD8^+^ T-cell activation. In contrast, semi-mature DCs exhibit low membrane CD40 levels and promote IL-10-producing regulatory T cells (Tregs) [5,6]. Unlike CD40, B7-H4 is a *negative* regulatory B7 family member that inhibits inflammatory CD4^+^ T-cell responses. Accordingly, B7-H4 is heavily expressed by aggressive tumor cells [7], and, in preclinical studies, the absence of B7-H4 is associated with severe GVHD [8]. Myeloid DCs (mDCs) were shown to mitigate GVHD by displaying antigens to T cells alongside regulatory co-stimulatory signals [9]; mDCs are divided into cDC2, mainly found in mucosal sites and in mesenteric lymph nodes, and cDC1, which reside mainly in the jejunum [10]. Compared to cDC1, cDC2 is a naive CD4^+^ T-cell stimulator and an inferior CD8^+^ T-cell antigen-presenter [11]. 

α1-Antitrypsin (AAT), the most abundant circulating serine protease inhibitor (0.9–1.75 mg/mL), is a tolerogenic tissue-protective molecule that expedites inflammatory resolution. AAT promotes semi-mature IL-10-producing DCs, expands Treg populations, and enhances anti-bacterial and anti-tumor NK cell responses [12,13,14]. The anti-proteolytic activity of AAT is counteracted by exposure to reactive oxygen species, and its overall activity is ablated by covalent binding to its target proteases. AAT is synthesized primarily by hepatocytes, and its levels rise up to sixfold during acute-phase responses [15]. At the genetic level, most individuals carry M alleles of AAT, which undergo changes in their glycosylation patterns during the resolution phase of systemic inflammation: from normative M2–M3 glycoforms to hyper-glycosylated M0–M1 glycoforms that possess enhanced immunomodulatory properties [16,17]. Individuals diagnosed with genetic AAT deficiency carry S, Z, or other non-M variants of AAT [18]; their levels of circulating AAT are below normal, and unless treated, they endure non-smoking lung tissue degradation and are susceptible to excessive inflammatory conditions [19]. Accordingly, they receive lifelong plasma-purified AAT augmentation therapy. 

At a biochemical level, AAT inhibits several serine proteases, most prominently neutrophil elastase, proteinase 3 (PR3), and cathepsin G [20]. Specifically, within the bone marrow niche, PR3 promotes IL-1-dependent inflammation [21] and affects hematopoietic stem cell phenotypes [22]. It is, therefore, not unexpected that loss of AAT via stool during gastrointestinal GVHD is predictive of poor response to corticosteroid therapy [23], and that AAT infusions improve clinical outcomes in patients with acute steroid-refractory GVHD [24,25,26,27]. Another acute-phase reactant, C-reactive protein (CRP), predicts poor prognosis in allogeneic HSCT [28,29]. However, the activities of CRP are distinct from those of AAT, and the protective functions of AAT throughout the multistep procedure of conditioning and subsequent HSCT are presently uncharted. While an apparent association between inactive AAT and poor HSCT outcomes was observed by Mullins RE et al. in as early as the year 1987 [30], it has yet to be determined whether insufficient AAT functionality *predisposes* allogeneic HSCT patients to acute GVHD, and in which time point relative to HSCT, at that.

In the present multicenter prospective study, blood samples were collected *before* and *after* patient conditioning and then throughout the course of HSCT, for measuring AAT and CRP levels, serum anti-PR3 activity, and immunomodulatory profiles of circulating patient immunocytes. The ex vivo response to clinical-grade AAT was tested on circulating patient leukocytes and on a human epithelial cell line treated with patient sera in a gap closure assay. 

## 2. Results

### 2.1. Patient Characteristics

The demographics, background disease, transplantation setting, and accompanying treatments of the 53 enrolled patients are summarized in Table 1. Their median age was 54 years (minimum to maximum: 24–74). The major regimen used was reduced-intensity conditioning (RIC), and the rest received myeloablative conditioning (MAC); 53% of patients received a matched unrelated donor graft. Overall, 26 patients (49%) developed acute GVHD (aGVHD), 16 (30%) developed mild (grade 1 or 2) aGVHD, and 10 patients (19%) developed severe aGVHD (grade 3 or 4). Five patients (9%) suffered from skin involvement alone, twelve (23%) suffered from gut involvement, and eight (15%) had gut and skin involvement. One patient suffered from liver involvement. Incidences of aGVHD appeared between 16 and 61 days from the time of HSCT (median: 34 days).

### 2.2. Circulating Levels of AAT and CRP and Serum Anti-PR3 Inhibitory Capacity before Conditioning and throughout the First 28 Days Post-HSCT

Patients exhibited a steady increase in circulating AAT levels, regardless of prospective GVHD status (Figure 1A). AAT levels peaked on day +14, while both serum PR3 inhibition capacity and CRP levels peaked on day +7 (Figure 1B,C). Of the 53 patients, 4 died before day +28; their data for the preceding time points were included with the rest of the cohort and excluded for GVHD status. 

Compared to GVHD-free patients, patients who developed aGVHD exhibited slightly higher AAT levels on days 0, +14, and +28, yet without reaching statistical significance (Figure 1D); PR3 inhibition was similarly superior on days 0, +7, and +14 (Figure 1E). CRP levels were low and uniform between the two groups, yet on day +14, mean CRP levels were low in GVHD-free patients (Figure 1F).

### 2.3. AAT Anti-PR3 Functionality Corrected to Inflammatory Status by CRP Pre- and Post-HSCT

In search of a quantifiable association between specific AAT functionality (PR3 inhibition/AAT) and the degree of non-specific inflammation (CRP), a three-pronged composite value was generated by the division of PR3 inhibition over AAT, then over CRP levels. At baseline, the composite value was superior to specific anti-PR3 functionality in predicting aGVHD (Figure 2A) and for varying grades of aGVHD (Figure 2B). To assess the prognostic value of the composite, cumulative incidences of aGVHD and NRM were modeled based on median values of AAT, anti-PR3 activity, the clinical cut-off of CRP, and the PR3/AAT/CRP ratio (Figure 2C,D). No single parameter predicted either aGVHD or NRM, while baseline PR3/AAT/CRP significantly correlated with aGVHD and NRM (Appendix A). To control for low-grade aGVHD (e.g., grade 1 or skin-only GVHD), a table was devised for the tested values, accordingly (Appendix A). Data from day +28 did not predict NRM (Appendix A).

### 2.4. Immunomodulation of Circulating Myeloid Dendritic Cells by Exogenous AAT

On day +28 post-transplantation, whole blood samples from 34 patients were evaluated ex vivo for cell subtypes in response to inflammatory stimulation and AAT augmentation. As shown (Figure 3A), there was no significant difference in cDC subsets between patient groups. Release of IL-6 and IL-10 from LPS- or R848–stimulated samples was unaffected by added AAT (Figure 3B). However, membrane expression of B7-H4 on both cDC subsets (Figure 3C,D) was significantly increased upon inflammatory stimuli (*colored bars*) and by the addition of AAT (*white bars*); in contrast, CD40 expression was overall unchanged by AAT. Between aGVHD and GVHD-free patients (Figure 3E), the cDC2 (CD1c^+^) subset displayed overall greater levels of B7-H4 expression in the GVHD-free group compared to the aGVHD group; a reciprocal relation was observed in cDC1 (CD141^+^) cells. Specifically, in the cDC2 subset, the GVHD-free group increased B7-H4 expression in response to added AAT, albeit without reaching statistical significance. A sharp rise was observed in R848-stimulated cDC1 cells from GVHD-free patients, irrespective of added AAT. A significant AAT-mediated rise in B7-H4 was observed in the cDC1 subset from patients that were below either or both the median of AAT concentrations and that of PR3/AAT/CRP values (Figure 3F). 

### 2.5. Colonic Epithelial Gap Closure and Anti-PR3 Serum Activity

The potential for tissue repair contained in serum samples was analyzed in vitro. Samples were collected on day +7 (representing tissue damage after transplantation) and on day +28 (imminent aGVHD). Sera from both aGVHD and GVHD-free patients modestly increased the relative wound density (RWD), and the addition of AAT alone to the 5% FBS control condition enhanced re-epithelialization rates (Appendix A). Adding AAT into patient sera from day +7 significantly improved re-epithelialization rates in the GVHD-free group, but not in the aGVHD group (ΔRWD 2.95% and −0.7%, respectively); in contrast, patient sera from day +28 exhibited a reciprocal trend (ΔRWD 0.19% and 1.44%, respectively) (Figure 4A). AAT augmentation of serum was also beneficial for patients with a low PR3/AAT/CRP composite value (Figure 4B). 

In search of a possible explanation for the change in AAT activity, the glycosylation patterns of AAT for each timepoint were determined (Figure 4C). As expected, M2/M3 glycosylation patterns (*black*) were abundant before conditioning (*pre-HSCT*) and on the day of transplantation (*day 0*), while hyper-glycosylated patterns (M1/M2 and M0/M1 bands, *gray* and *blue*, respectively) were abundant after HSCT, comprising the majority of cases on day +14. Serum samples from day +7, containing the hyper-glycosylated M0/M1 glycoforms, modestly improved epithelial gap repair (Figure 4D). 

## 3. Discussion

Current clinical trials examine AAT therapy for the prevention of GVHD (NCT03805789) and in combination with corticosteroids for the treatment of acute GVHD (NCT04167514). The present study set out to explore several aspects of *endogenous* circulating AAT as a dynamic biomarker for the risk of developing acute GVHD, with relevance to baseline tissue damage and systemic inflammation, both detrimental to HSCT outcomes. Marcondes AM et al. showed that low serum AAT levels in BM donors are associated with subsequent GVHD severity [31], and it was carefully suggested that donor leukocytes may carry tolerogenic immune activation profiles when extracted from an AAT-rich environment. It is suggested that, by inhibiting PR3, AAT may regulate the bone-marrow-localized protease–antiprotease relationship, suppressing, e.g., IL-32 activity [32] and reducing neutrophil membrane PR3 levels [33]. By this, myeloid maturation may be diverted towards less alloreactive immune cells [34]. 

Expectedly, circulating AAT levels alone did not predict the occurrence of acute GVHD. However, linking its concentrations with early changes in anti-PR3 activity that is contained within the serum, and with CRP that represents nonspecific inflammatory status and widespread tissue injury, did predict the occurrence of acute GVHD, perhaps signifying early failure of immunoregulation in the face of underlying widespread tissue damage. Of note, while MAC is expectedly more tissue damaging than RIC, there were no statistically significant differences between groups across the tested parameters, possibly owing to the particular rationale for each conditioning approach afforded per recipient; an interesting trend in the myeloablative conditioning group appeared in the form of somewhat higher PR3/AAT/CRP ratio values on day 0 (Appendix A), although larger trials are required to ascertain whether this trend reaches statistical significance. Moreover, the fact that production of AAT by liver cells is affected by standard immunosuppressive agents may pose as a limiting factor in achieving a GVHD-free clinical sequel.

CRP and AAT are both liver products triggered by inflammation and hypoxia, yet their half-life differs (19 h and 3–5 days, respectively) [15,35], as does their function. As endothelial damage plays an inherent role in HSCT [36], it is postulated that AAT augmentation may minimize the development of veno-occlusive disease [37]. In the present study, however, patients with veno-occlusive disease were not followed since three of the four centers taking part in the trial were conducting an ongoing study involving pre-emptive treatment for veno-occlusive disease. Additionally, unlike CRP, AAT increases IL-1Ra production [38] and therefore diminishes IL-1-dependent IL-6 production. Here, the novel observation by which changes in AAT levels appear to *lag* behind changes in CRP levels points to the importance of combining parameters of tissue damage and variables that signify systemic inflammation, for better interpretation of underlying biological processes. Indeed, the phenomenon of *relative* insufficiency in AAT functionality is supported by studies that depict high CRP levels accompanied by *inadequate* AAT responses in severe COVID-19 patients [39,40].

Increased expression of B7-H4 by AAT on both subsets of cDCs is in agreement with AAT interfering with the CD40:CD40L pathway, as with the B7.1 (CD80) and B7.2 (CD86) pathways [12,31,41]. The clinical use of checkpoint inhibitors prior to allo-HSCT, as well as low soluble programmed cell death protein 1 (PD-1) levels, is associated with high rates of GVHD [42,43]. In the present study, cDC2 cells from GVHD-free patients exhibited higher B7-H4 in response to added AAT in vitro, implying some tolerogenic potential for AAT augmentation. Furthermore, resolution of possible AAT insufficiency is suggested by the response of increased B7-H4 expression in patients with low AAT concentrations or low PR3/AAT/CRP value. These findings should be reviewed carefully when considering the *prevention* of GVHD, due to the relative risk of interfering with the graft-versus-leukemia effect, although in vivo studies depict AAT as favoring uninterrupted anti-tumor NK cell responses [14,41].

In studying the biology of AAT, it is important to address not only its levels and anti-protease activity, but also its molecular architecture; being a plasma protein, glycosylation branches embedded in its structure are responsive to inflammatory conditions. The emergence of increased glycosylation patterns in circulating AAT, parallel to changes in CRP, entails a physiological liver response to conditioning-related tissue damage, as suggested elsewhere [44]. Improved epithelial gap closure capacity by day +7 sera containing M0/M1-hyperglycosylated AAT supports the relevance of AAT functionality during the resolution phase. 

In a recent study, biomarker-guided preemptive AAT therapy was evaluated in high-risk patients on days +7 and +14 post-HSCT; nonetheless, the study found no change in incidences of steroid-refractory GVHD [45]. This may be the result of either an insufficient dose of AAT or a relatively late intervention with AAT. According to animal studies, optimal AAT supplementation includes treating *donors* before bone marrow isolation and recipients *before* transplantation [31,38]. The controversy around patient selection for AAT therapy prior to conditioning arises from the notion that it may have an adverse effect on the graft-versus-tumor response, even though preclinical studies suggest that anti-tumor NK cell activities are enhanced by AAT [14]. Levin et al. demonstrated that initiating AAT therapy as early as day +7 for high-risk patients, based on the MAGIC biomarkers criteria, did not provide a benefit in preventing acute GVHD [45]. However, a retrospective study that evaluates these high-risk patients at their baseline and considers their disease status before conditioning as an indicator for selective treatment with AAT could potentially help control tissue damage and minimize the undesirable effects of GVH response. This can be accomplished by administering AAT to either patients or donors, as well as hematopoietic stem cells. Combinations of AAT with other immunosuppressants should be explored, as AAT alone did not modulate IL-6 and IL-10 levels in whole blood assays under the conditions explored in the present study. 

Although statistically underpowered, it was important to study the biology of circulating AAT in a longitudinal manner. Nonetheless, knowingly, the requirement for multiple HSCT medical centers, necessary for reaching a reasonable cohort size, added a large degree of variability. For instance, in one center, following conditioning, seven patients were treated with plasma infusions twice a week; their circulating AAT concentrations were not affected, yet their total PR3 inhibition capacity was significantly lower compared to the rest of the cohort (Appendix A). 

In conclusion, the present study explored translational aspects relating to acute GVHD and the biology of endogenous AAT in the context of patient conditioning prior to HSCT. It is suggested that enhanced tissue repair capacity and early modulation of APCs, both well-established attributes of AAT, may influence the brittle immunoregulation that accompanies HSCT patients, and the subsequent development of acute GVHD. These prospects should be further challenged and validated in larger clinical trials. 

## 4. Materials and Methods

### 4.1. Study Approval and Design

This study was carried out in four clinical centers in Israel: Rambam Health Care Campus, Tel-Aviv Sourasky Medical Center, Hadassah Medical Center, and Soroka University Medical Center. The study was approved by the respective institutional review boards (0196-17-RMB, 0370-17-TLV, 0402-17-HMO, and 0404-18-SOR). It is registered at ClinicalTrials.gov (identifier NCT03188601) and the Israeli Ministry of Health website (MOH_2017-08-10_000573). 

Between 2017 and 2020, 53 adult patients scheduled for allo-HSCT were recruited. Written informed consent was obtained. All patients were observed for at least 1 year. Blood samples for serum analysis were collected on the day of conditioning initiation, as well as immediately before stem cell infusion (day 0) and then on days +7, +14, and +28 (with a 3-day deviation margin). Blood samples for whole blood assays and for peripheral blood mononuclear cells (PBMCs) were obtained on the day of conditioning initiation and on day +28 in sterile lithium heparin tubes (VACUETTE^®^, Greiner Bio-One, Kremsmünster, Austria).

### 4.2. Quantification and Enzymatic Assays

Sera were tested for AAT and CRP levels by means of immunoassay using a Roche Cobas^©^ 6000, Indianapolis, IN, USA and a Beckman Coulter^©^ AU Analyzer 5400, Brea, CA, USA. The activity of human neutrophil PR3 (Athens Research & Technology, Inc., Athens, GA, USA) was determined by means of fluorescence resonance energy transfer (FRET) assay using 5-TAMRA-VAD-Nva-RDYQ-Dap(5-CF) (EMC Microcollections, Tübingen, Germany), described elsewhere [46]. Briefly, sera were diluted ×100 and mixed with PR3 (10 nM) in activity buffer (50 mM Tris, 150 mM NaCl, and 0.01% Triton X-100, pH 7.4), then incubated in duplicate for 20 min at 37 °C. The reaction was monitored for 1 h at Ex/Em 485/530 nm directly after adding 2.5 µM FRET substrate, and relative activity was deduced after correction for the negative control and against the control PR3 kinetic slope.

### 4.3. Whole Blood and Peripheral Blood Mononuclear Cell Activation Assay and Flow Cytometry Analysis

Peripheral blood samples (18 mL per time point) were used for whole blood analysis and PBMC isolation. Whole blood assay samples were diluted 1:1 in preheated 0% FBS RPMI medium (Biological Industries, Beit-Haemek, Israel). Whole blood samples and PBMCs were incubated in capped 5 mL polypropylene tubes and treated for 1 h with 0.5 mg/mL clinical-grade human AAT (Glassia^©^; KAMADA, Beit-Kama, Israel) followed by stimulation with either 10 ng/mL LPS (*Escherichia coli* O55:B5; Sigma-Aldrich, St. Louis, MO, USA) or 100 ng/mL R-848 (Resiquimoid, Caymanchem, MI, USA). Supernatants were collected after 4 h for cytokine analysis. For cytometric analysis, mDCs were defined as previously reported [47]. PBMCs were separated by using Histopaque^®^-1077 (Sigma-Aldrich, MO, US GE Healthcare Life Sciences, Marlborough, MA, USA) in 1:1 PBS-diluted blood; mononuclear cells were aspirated before treatment with red blood cell lysis buffer (Ammonium-Chloride-Potassium Lysis Buffer). Cells were resuspended in FACS buffer (1% BSA, 0.1% sodium azide, and 2 mM EDTA in PBS, all from Sigma-Aldrich). Blocking was performed prior to staining using Tandem Signal Enhancer (Miltenyi Biotec, cat. 130-099-887, Bergisch Gladbach, Germany) at room temperature for 20 min, then cells were stained with antibodies purchased from Miltenyi Biotec for CD1c (clone REA694, cat. 130-110-536), CD11c (clone REA618, cat. 130-114-584), CD14 (clone REA599, cat. 130-110-523), CD141 (clone REA674, cat. 130-110-928), CD127 (clone REA614, cat. 130-109-518), CD25 (clone REA570, cat. 130-109-076), CD3 (clone REA613, cat. 130-110-464), CD366-TIM3 (clone REA635, cat. 130-109-759), CD4 (clone REA623, cat. 130-110-950), CD40 (clone REA733, cat. 130-110-950), CD8 (clone REA734, cat. 130-110-820), FoxP3 (clone 3G3, cat. 130-093-014), and HLA-DR (clone REA805, cat. 130-111-792). Antibody to B7-H4 was purchased from Biolegend (clone MIH43, cat. 358106). Cells were classified as myeloid DCs, then subclassified into cDC1 (CD14^-^CD11c^+^HLA-DR^+^CD141^+^) or cDC2 (CD14^-^CD11c^+^HLA-DR^+^CD1c^+^), and then analyzed for CD40 and B7-H4 membrane expression (Appendix A). Data were acquired using a BD-FACSAria-III (BD Biosciences, CA, USA) and analyzed using FlowJo™ Software (Windows) version 10.7.1. Ashland, OR, USA: Becton, Dickinson and Company.

### 4.4. Epithelial Gap Closure Assay

The human epithelial colorectal adenocarcinoma cell line Caco2BBE1 (ATCC, Manassas, VA, USA) was kindly provided by Dr. Ehud Ohana (Ben-Gurion University of the Negev, Israel). Cells were maintained in DMEM (10% FBS, 2 mM L-glutamine, 100 U/mL penicillin/streptomycin, 1 mM sodium pyruvate, and human transferrin, all from Biological Industries). The medium was replaced every 2 days, and cells were subcultured every 3–4 days using trypsin-EDTA (Biological Industries). Cells were grown to confluence in 96-well plates (8 × 10^4^ cells/well), were treated for 2 h with mitomycin C (5 µg/mL; Sigma-Aldrich), and then underwent scratching using Incucyte^©^ S3 Woundmaker (Sartorius Corporation, Ann Arbor, MG, USA) before being treated under the indicated conditions in 5% FBS medium. Images were acquired twice per well every 6 h for 24 h and analyzed using Incucyte^©^ S3 Woundmaker. The change from the initial wound area was defined as the percent relative wound density (RWD).

### 4.5. Isoelectric Focusing and Phenotyping

AAT protein phenotyping and glycosylation analysis in patient sera were performed by means of the immunofixation of serum glycoforms via isoelectric focusing gel electrophoresis using HYDRASYS (Sebia, Lisses, France). Phenotypes were confirmed by means of paired allele genotyping (Appendix A).

### 4.6. Statistical Analysis

Descriptive statistics are presented as the mean ± standard error of the mean (SEM) and as the median and interquartile range (IQR) for quantitative non-normally distributed variables. The numbers of cases and percentages were computed based on all available cases for categorical variables. To compare multiple groups, one- and two-way ANOVA were used with the Dunn–Sidàk post hoc test. Differences in mRNA expression were determined using the Mann–Whitney test. The cumulative incidence of NRM was calculated with relapse as a competing risk. Gray’s test was used to compare the cumulative incidence function (CIF) between groups. Multivariable linear mixed models with the patient as a random intercept were used for repeated measure analysis [48]. Data were analyzed using Windows GraphPad Prism version 9.0.2 (GraphPad Software, La Jolla, CA, USA) and R studio “lmer” and “cmprsk” packages. Results were considered statistically significant at *p* < 0.05.

## Figures and Tables

**Figure 1 ijms-25-00422-f001:**
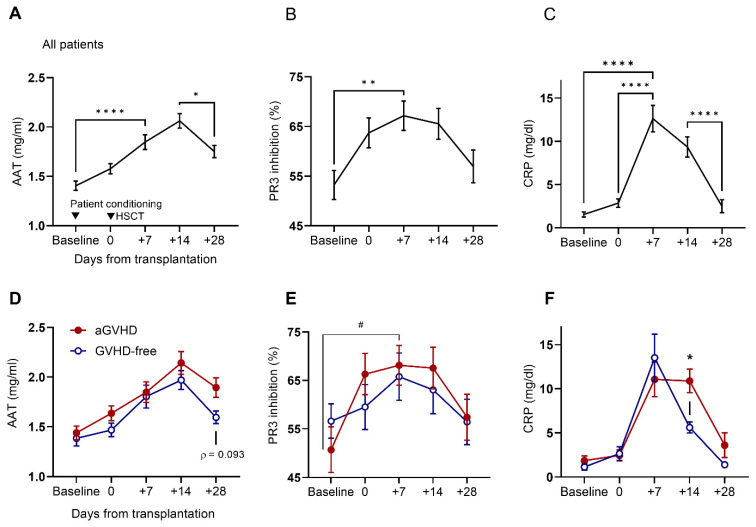
Levels of AAT, PR3 inhibition, and CRP increase early after transplantation and with eventual GVHD diagnosis. Temporal distribution of (**A**) AAT levels, (**B**) PR3 inhibition capacity, and (**C**) CRP levels in sera collected from 53 patients (n = 48–52 per timepoint). Kruskal–Wallis multiple comparisons test. Mean ± SEM. Distribution by individuals who developed acute GVHD (aGVHD, *solid circle*, n = 26) and individuals that were GVHD-free (*open circle*, n = 23) throughout the follow-up period for (**D**) AAT levels, (**E**) PR3 inhibition capacity, and (**F**) CRP levels. Two-way ANOVA with Sidák’s post hoc test. Mean ± SEM. * Comparison between aGVHD and aGVHD-free, # comparison within patient group. For * and #, * *p* < 0.05, ** *p* < 0.01, **** *p* < 0.0001.

**Figure 2 ijms-25-00422-f002:**
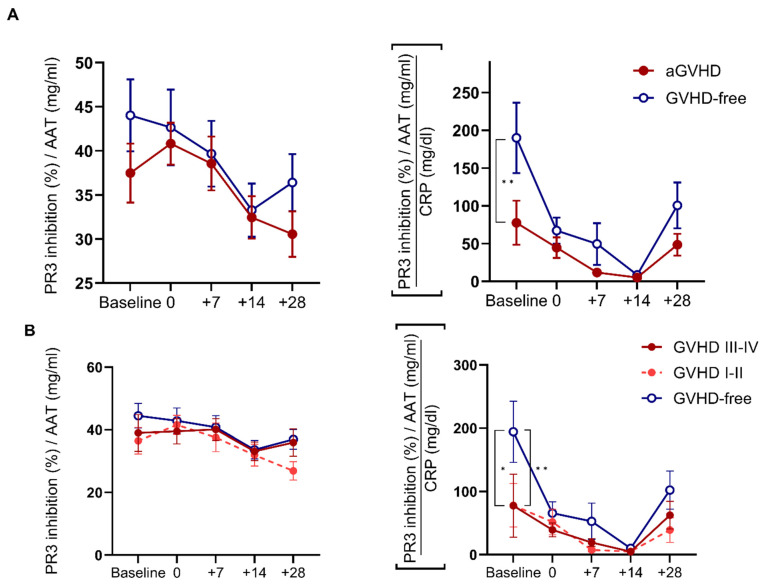
Combined parameters for the prediction of aGVHD and NRM according to pre-HSCT baseline values. (**A**) Acute GVHD diagnosis according to specific PR3 inhibition (PR3 inhibition divided by AAT levels) and a triple-parameter numerical score (calculated ratio between serum PR3 inhibition and AAT, divided by CRP levels). Two-way ANOVA with Sidák’s post hoc test. Mean ± SEM. (**B**) Specific PR3 inhibition and a triple-parameter numerical score according to aGVHD grade; GVHD-free throughout the follow-up period (*open circle*, n = 23), aGVHD grade I-II (*dashed line*, n = 16), aGVHD grade II-IV (*solid square*, n = 10). Two-way ANOVA with Sidák’s post hoc test. Mean ± SEM. The probability of developing (**C**) aGVHD and (**D**) NRM from the PR3/AAT/CRP ratio at the time of preconditioning. Gray’s tests with relapse as competing risks. * Comparison between aGVHD and aGVHD-free. For *, * *p* < 0.05, ** *p* < 0.01.

**Figure 3 ijms-25-00422-f003:**
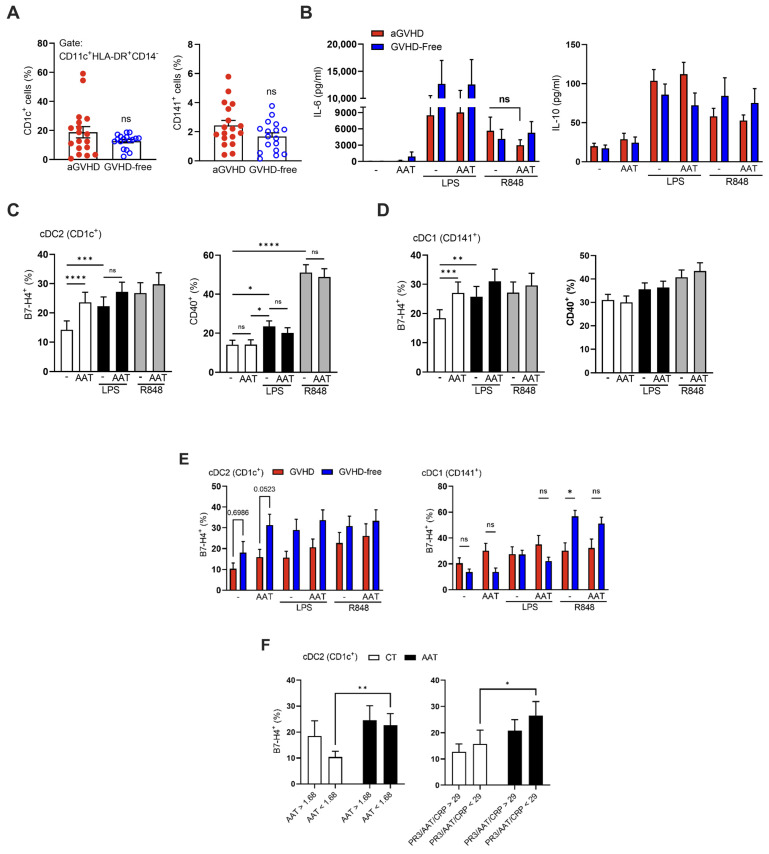
The effect of clinical-grade AAT on patient-derived myeloid dendritic cells. (**A**) Peripheral blood samples from aGVHD patients (n = 18) and GVHD-free (n = 17) patients. Percent conventional dendritic cells of type 1 (CD141^+^) and type 2 (CD1c^+^) out of CD14^-^CD11C^+^HLA-DR^+^ cells. (**B**–**E**) Whole blood and PBMC stimulation assays: samples from day +28 incubated with LPS (10 ng/mL) or R848 (100 ng/mL), ±AAT (0.5 mg/mL). (**B**) IL-6 and IL-10 supernatant levels. aGVHD (n = 17) and GVHD-free (n = 15–16). Two-way ANOVA with Sidák’s post hoc test. Mean ± SEM. (**C**,**D**) Membranal expression by flow cytometric analysis of B7-H4 and CD40 on cDC1 and cDC2 (n = 34). (**E**) Membranal expression of B7-H4 by flow cytometric analysis according to disease status: aGVHD (n = 17) or GVHD-free (n = 17), and (**F**) according to median cutoffs: median AAT concentration (1.68 mg/mL, n = 16, 18) or median PR3/AAT/CRP (29 AU, n = 17, 17). Two-way ANOVA with Sidák’s post hoc test. Mean ± SEM. * *p* < 0.05, ** *p* < 0.01, *** *p* < 0.001, **** *p* < 0.0001.

**Figure 4 ijms-25-00422-f004:**
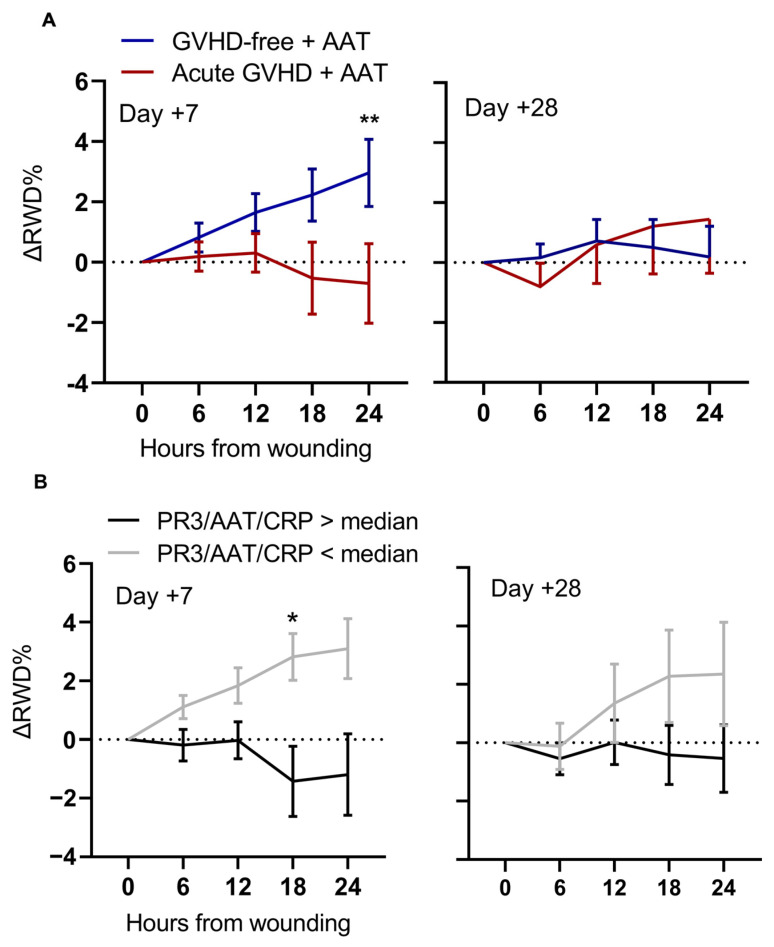
The effect of AAT glycosylation patterns and exogenous AAT on serum capacity to promote re-epithelialization of intestinal epithelial cells. (**A**) Change in relative wound density (ΔRWD) in Caco2-BBe1 cell monolayers treated with patient serum supplemented with 10 µg/mL AAT, according to aGVHD and GVHD-free groups (n = 22–24 per group on designated days). (**B**) ΔRWD distribution by PR3/AAT/CRP median cutoff (3.4 and 36.2, median, for days +7 and +28, respectively). (**C**) Glycosylation patterns of circulating AAT (n = 46–50): normal glycosylation (M2–M8 bands), increased glycosylation (M1 + M2 bands), and extreme glycosylation (M0 + M1 bands). (**D**) Relative wound density (RWD) in serum-treated C2BBE1 cells by glycosylation patterns. Samples from day +7. Normal glycosylation (thick line, n = 21); M1 + M2 (dashed line, n = 23–24); M0 + M1 (dotted line, n = 5). Two-way ANOVA with Sidák’s post hoc test. Mean ± SEM. * *p* < 0.05, ** *p* < 0.01.

**Table 1 ijms-25-00422-t001:** Patient characteristics.

Cohort Characteristics (N = 53)	Median (Range)
Age (yrs)	54 (24–74)
Male sex (%)	30 (56.6)
**Indication for HSCT**	N (percent)
AML	25 (13.2)
ALL	8 (47.2)
Non-Hodgkin’s lymphoma	7 (15.1)
MDS	6 (13.2)
Other *	7 (11.3)
**Donor characteristics**	Median (range)
Age (yrs)	27 (13–71)
Male sex	42 (79.2%)
Related	24 (45.3%)
Unrelated	28 (52.8%)
Haploidentical	1 (1.9%)
Female to male	5 (9.4%)
**HLA-match**	N (percent)
Matched	49 (92.4)
Mismatched	3 (5.7)
Haploidentical	1 (1.9)
**Conditioning regimen intensity**	N (percent)
Myeloablative conditioning	19 (35.8)
Reduced-intensity conditioning	34 (64.2)
**GVHD prophylaxis**	N (percent)
MTX + CNI	32 (60.4)
MMF + CNI	17 (32)
MTX + CNI + MMF	3 (5.7)
RAP + CNI	1 (1.9)
ATG	33 (62.2)

AML, acute myeloid leukemia; ALL, acute lymphoblastic leukemia; MDS, myeloid dysplastic syndrome; MTX, methotrexate; CNI, calcineurin inhibitor; MMF, mycophenolate mofetil; RAP, rapamycin; ATG, anti-thymocyte globulin. * Other: primary myelofibrosis, congenital neutropenia, multiple myeloma, T-cell prolymphocytic leukemia, mixed-phenotype acute leukemia, and chronic myeloid leukemia.

## Data Availability

All data generated or analyzed during this study are included in this published article and its Appendix A.

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
