# Peer review of "Altered Serum Alpha1-Antitrypsin Protease Inhibition before and after Clinical Hematopoietic Stem Cell Transplantation: Association with Risk for Non-Relapse Mortality"

_ijms, 2023, doi:10.3390/ijms25010422_

Round 1
Reviewer 1 Report
Comments and Suggestions for Authors
The Authors have reported on a study regarding anti-proteolytic function of circulating AAT in the context of allogeneic stem cell transplantation.
The manuscript is well designed and readable, despite the complex physiological and pathological interplay of markers described.
I have only a few comments or minor changes to suggest:
1) in the abstract and in the main text, allo-HSCT is defined as a potentially curative intervention for leukemia, but it should be noted that it has many more indications; I would change leukemia with hematological diseases.
2) Myeloablative conditioning regimens are referred to as “Cytotoxic ablation protocols” which might be vague; actually, among patients enrolled, most frequently employed conditioning regimens were RIC. I also wonder whether the Authors have checked if there is substantial difference in AAT/PR3/CRP levels between RIC and MAC groups; however I feel it should be worth adding it to the manuscript, since tissue damage after MAC conditioning is clearly higher.
3) page 4 row 187 and in the following pages: it should read “p <0.05” and not “rho”
4) Since VOD is cited in the discussion but not in the results, I suggest to add the incidence of veno-occlusive disease in enrolled patients in the manuscript.
Author Response
Reviewer #1
The Authors have reported on a study regarding anti-proteolytic function of circulating AAT in the context of allogeneic stem cell transplantation. The manuscript is well designed and readable, despite the complex physiological and pathological interplay of markers described. I have only a few comments or minor changes to suggest:
Reviewer: in the abstract and in the main text, allo-HSCT is defined as a potentially curative intervention for leukemia, but it should be noted that it has many more indications; I would change leukemia with hematological diseases.
Authors: Point well accepted and corrected.
Reviewer: Myeloablative conditioning regimens are referred to as "Cytotoxic ablation protocols" which might be vague; actually, among patients enrolled, most frequently employed conditioning regimens were RIC. I also wonder whether the Authors have checked if there is substantial difference in AAT/PR3/CRP levels between RIC and MAC groups; however I feel it should be worth adding it to the manuscript, since tissue damage after MAC conditioning is clearly higher.
Authors: This is indeed an important note. There were no substantial differences between either myeloablative conditioning or matched unrelated donors as harbingers of excessive tissue damage, although this was highly anticipated by both researchers and clinicians taking part in this study. However, following reviewer’s suggestion, we dedicated an extra panel to Figure S6 with the text:
“Of note, while MAC is expectedly more tissue damaging than RIC, there were no statistically significant differences between groups across tested parameters, possibly owing to the particular rationale for each conditioning approach afforded per recipient; an interesting trend in the myeloablative conditioning group appeared in the form of somewhat higher levels of PR3/AAT/CRP ratio on day 0 (Supp. Fig. 6), although larger trials are required to ascertain whether this trend reaches statistical significance.”
Reviewer: page 4 row 187 and in the following pages: it should read "p <0.05" and not "rho"
Authors: Point well accepted and corrected.
Reviewer: Since VOD is cited in the discussion but not in the results, I suggest to add the incidence of veno-occlusive disease in enrolled patients in the manuscript.
Authors: This is indeed an important note. We did not follow patients with veno-occlusive disease in our study since 3 of the 4 centers taking part in our trial had a parallel study with pre-emptive treatment for VOD. This piece of information has been added to the Discussion section:
“In the present study, however, patients with veno-occlusive disease were not followed since 3 of the 4 centers taking part in the trial had an ongoing study with pre-emptive treatment for veno-occlusive disease.”
Reviewer 2 Report
Comments and Suggestions for Authors
Well written and agree with the need for a larger study
What would be nice, would be a comment or two on prophylactic and/or therapeutic options that might be significant beyond what they describe in animals.
Specifics - what is the breakdown of related vs unrelated donors and is there a difference even though not statistically found.
Is there any difference between MA and RIT conditioning
is there any
Author Response
Reviewer #2
Well written and agree with the need for a larger study. What would be nice, would be a comment or two on prophylactic and/or therapeutic options that might be significant beyond what they describe in animals.
Authors: This is a great notion that we believe is crucial in understanding the mechanisms by which alpha-1-antitrypsin (AAT) exerts its tissue protective effects and some of it is described in the discussion. The section is now added some more insights:
The controversy around patient selection for AAT therapy prior to conditioning arises from the notion that it may have an adverse effect on the graft versus tumor response, even though preclinical studies suggest that anti-tumor NK cell activities are enhanced by AAT (28003813). Levin et al. demonstrated that initiating AAT therapy as early as day +7 for high-risk patients, based on the MAGIC biomarkers criteria, did not provide a benefit in preventing acute GVHD (33351103). However, a retrospective study that evaluates these high-risk patients at their baseline and considers their disease status before conditioning as an indicator for selective treatment with AAT, could potentially help control tissue damage and minimize the undesirable effects of GVH response. This can be accomplished by either administering AAT to patients or donors, as well as the hematopoietic stem cells.
Reviewer: Specifics - what is the breakdown of related vs unrelated donors and is there a difference even though not statistically found. Is there any difference between MA and RIT conditioning
Authors: As described in response to Reviewer 1. This is indeed an important note. There were no substantial differences between either myeloablative conditioning or matched unrelated donors as harbingers of excessive tissue damage, although this was highly anticipated by both researchers and clinicians taking part in this study. However, following reviewer’s suggestion, we dedicated an extra panel to Figure S6 with the text:
“Of note, while MAC is expectedly more tissue damaging than RIC, there were no statistically significant differences between groups across tested parameters, possibly owing to the particular rationale for each conditioning approach afforded per recipient; an interesting trend in the myeloablative conditioning group appeared in the form of somewhat higher levels of PR3/AAT/CRP ratio on day 0 (Supp. Fig. 6), although larger trials are required to ascertain whether this trend reaches statistical significance.”